# The Effects of the Transmigration Programme on Poverty Reduction in Indonesia's Gorontalo Province: A Multidimensional Approach

Amelia Murtisari [1,2], Irham Irham [3,*], Jangkung Handoyo Mulyo [3] and Lestari Rahayu Waluyati [3]

1    Doctoral Study Program, Faculty of Agriculture, Universitas Gadjah Mada, P.O. Box 16, Bulaksumur, Yogyakarta 55281, Indonesia
2    Department of Agribusiness, Faculty of Agriculture, Universitas Negeri Gorontalo, Gorontalo 96122, Indonesia
3    Department of Agricultural Socioeconomics, Faculty of Agriculture, Universitas Gadjah Mada, P.O. Box 16, Bulaksumur, Yogyakarta 55281, Indonesia
*    Correspondence: irham@ugm.ac.id; Tel.: +62-812-2721-5348

**Abstract:** The transmigration program in Gorontalo Province plays an important role in poverty reduction. The districts of Gorontalo and Boalemo, as a part of Gorontalo Province, were purposefully selected as research locations. A total of 240 respondents of transmigrant and local households were interviewed using a questionnaire containing a list of questions with a 5-Likert scale. The objectives of this research are: (1) to measure the multidimensional poverty level of transmigrant and local households, (2) to determine the impact of the transmigration program on poverty reduction and (3) to identify the factors that influence poverty status of transmigrant and local households. The results show that the poverty level of transmigrant households tends to be lower than local households. The longer the placement of transmigration, the more likely it is to reduce regional poverty levels. The health dimension has a high contribution to the cause of poverty of transmigrants, while the education dimension contributes to the highest cause of poverty of local households. The results of the analysis also show that farmers who are more educated, participate in skills training and have a side business have more opportunity to reduce poverty. The study confirms that the transmigration program has a significant impact on poverty reduction in the region.

**Keywords:** transmigration program; transmigrant; local; poverty reduction; multidimensional poverty

## 1. Introduction

The transmigration program is one of the programs of the Ministry of Villages, Development of Disadvantaged Regions and Transmigration of the Republic of Indonesia, by voluntarily moving people from densely populated areas to sparsely populated areas. In the transmigration area, an agricultural zone is provided for the transmigrants that functions as a settlement and farming business for the community. Therefore, every transmigration household gets a yard, farmland 1 and farmland 2.

The transmigration program plays an important role to alter various dimensions of life, including forming a new socio-cultural identity for the community as well as patterns of economic development (Nova 2016), the assimilation of arts and improving welfare (Septiyani 2014). This is reinforced by research (Setyorini et al. 2018), which explains that regional development is supported by the transmigration program in terms of progress in infrastructure, economic growth and the creation of optimal production centres. Therefore, it is interesting to further study the role of the transmigration program in reducing poverty.

Poverty is still a serious problem due to the fact that there are 15.61% or 186.29 thousand poor people in Gorontalo Province (Badan Pusat Statistik Provinsi Gorontalo 2022), most of whom are scattered in rural areas. Figure 1 explains that, based on data for

the last five years, the poverty rate in rural areas is much higher than in urban areas, with a gap of more than 19% between them. Rural communities who rely on the agricultural sector as the main source of family income tend to have high business risks due to crop failure and low productivity due to inadequate rainfall, global climate change, the dangers of plant pests and diseases, land degradation, poor infrastructure, inadequate land size, which is narrow due to land conversion, and unequal food distribution (Purwaningsih 2008; Santoso and Nurumudin 2020). Not only that, farmers' limitations in accessing financial resources, information and access to markets, as well as the failure to transform agricultural technology, further exacerbate rural poverty rates (Rosyadi 2017; Possumah et al. 2018).

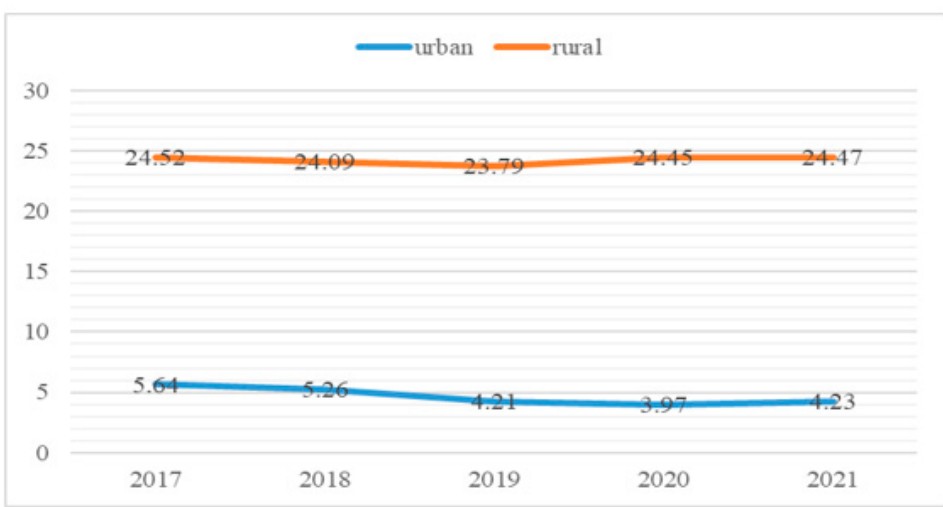

**Figure 1.** Percentage of rural and urban poverty rates in Gorontalo Province 2021.

The measurement of poverty is generally carried out based on the monetary aspect, mostly seen from the decline in income or consumption expenditure (Wisor 2012). However, monetary measurement is considered inaccurate because every individual who has sufficient income to meet basic needs does not necessarily use his income to spend (Thorbecke 2007). Therefore, the measurement of poverty is completed by involving non-monetary aspects in several studies known as multidimensional measurement (Multidimensional Poverty Index) (Alkire and Foster 2011b; Moonansingh et al. 2019; Bader et al. 2016; Djossou et al. 2017).

In measuring the Multidimensional Poverty Index (MPI) with the Alkire–Foster method approach, there are at least three measurement dimensions used, namely education, health and living standards (Kahlan et al. 2021). Several studies on measuring poverty have been developed, starting by expanding the dimensions of measurement to adding indicators for measuring poverty. The new thing that is shown in this research is the inclusion of new indicators on the dimensions of education, health and living conditions. Indicators of school participation during the COVID-19 pandemic, skills training and level of knowledge (education dimension), participation in social health insurance (health dimension) and farm condition (living condition) were included in this study to provide research novelties that had never been addressed in previous research. Comparing transmigrant and local households is also another new perspective, since they are unique in terms of regional origin and different cultural backgrounds.

Several studies use the education dimension as a variable for measuring poverty, including (Shaukat et al. 2019), by using one indicator, namely the level of education from no education to primary, secondary and higher. In the study, (Sumargo and Simanjuntak 2019; Datt 2018) included indicators of school years and school attendance in measuring the dimensions of education. Likewise in research (Haris 2016), indicators of school participation for children aged 7–18 years and preschool education for children aged 5–6 years are used for measuring the dimensions of education. The things that distinguish this research

from previous ones are the inclusion of new indicators, namely school activities during the COVID-19 pandemic, participation in skills training and ownership of agricultural and non-agricultural skills. Those indicators are needed because education is not only formal but also non-formal through training activities.

The next dimension is health. Indicators of nutrition and the number of child deaths are used as indicators for measuring the dimensions of health (Alkire et al. 2021). Different conditions were shown in the study (Lu et al. 2018), which measured the health dimension by using indicators of the health of family members and the use of the nearest health facility. Different things were analysed in research (Kahlan et al. 2021), which used three indicators, including child mortality, people with disabilities and mental health conditions. The novelty of the health dimension in this study includes indicators of participation in health insurance and the ability to pay for health facilities. These two indicators are included in the analysis because the Indonesian government is promoting health insurance through the services of the Social Security Administering Body (called BPJS).

Livelihood conditions are the third dimension that is included in the measurement of multidimensional poverty. In the study (Haris 2016), indicators of sanitation, access to lighting sources, house assets and the condition of the roof, floor and walls of the house were used as indicators for measuring livelihood. Other indicators are shown in the study (Alkire et al. 2021), such as the use of cooking fuel, drinking water, housing, sanitation, electricity and assets. The novelty of the livelihood dimension in this study is an indicator of livestock ownership and road conditions because the majority of transmigrant and local households own livestock as a source of family income. The condition of the road was chosen because the transmigrant area needs to have good supporting infrastructure such as roads. In addition, road facilities are used as access to carry out economic activities. Another novelty offered in this study is to compare transmigrant and local households with diverse social and cultural characteristics.

Based on various reviews of the above problems supported by several theoretical foundations, the research objectives include: (1) measuring the poverty level based on a multidimensional approach, (2) knowing the impact of the transmigration program on poverty reduction by comparing transmigrant and local households and (3) identifying the factors that affect the poverty status of transmigrant and local households.

## 2. Methodology

Survey method was used in this study by conducting interviews using a questionnaire containing a list of questions related to poverty measurement. The selection of research locations was carried out purposefully covering the districts of Gorontalo and Boalemo as the location of the transmigration program and included the districts with the highest poverty rates in Gorontalo Province. The research locations selected include Bongo I and II, Ayumolingo, Bukit Aren and Pangea SP3 Saritani. The research locations were divided into 3 groups based on the year of placement, including initial placement groups (Bongo I and II consisting of Harapan and Raharja villages), medieval placement (Ayumolingo and Bukit Aren villages, which included Ayumolingo and Bukit Aren villages) and new placements (Pangea SP 3 located in the villages of Saritani).

The sampling technique using random sampling method was used for each transmigrant and local household population. From each population was taken 30 respondents; according to (Sugiyono 2012; Gay et al. 2009), this number is considered feasible for quantitative research. Based on Table 1, the number of research respondents was 240 households consisting of transmigrants and locals.

**Table 1.** Research Respondents.

| Transmigration Placement Location | Placement Year | Village | Respondent Type | | Total |
|---|---|---|---|---|---|
| | | | Transmigrant | Local | |
| Bongo I and II | Initial (1976) | Harapan and Raharja | 30 | 30 | 60 |
| Ayumolingo and Bukit Aren | Medieval (2016) | Ayumolingo and Bukit Aren | 60 | 60 | 120 |
| Pangea SP 3 | New (2019) | Saritani | 30 | 30 | 60 |
| | Total | | 120 | 120 | 240 |

Note: local respondents are original people of Gorontalo Province, transmigrants are people from outside Gorontalo Province.

Based on the Alkire–Foster method (Alkire and Foster 2011a; Lu et al. 2018; Alkire et al. 2016), this study used three dimensions in measuring multidimensional poverty (Table 2), namely education, health and living conditions, with 23 indicators developed according to the characteristics of the research location. The multidimensional poverty analysis was carried out in two stages: the first was to determine the dimensions, indicators and thresholds, and the second was to explain the construction of these measures. Details of indicators for each dimension are described as follows:

1.  Education

The education dimension consists of eight indicators, including (1) the head of the family's time to finish elementary school, (2) the head of the family's junior high school education, (3) the presence of children who are not in school in a farmer's household, (4) children's school participation during the COVID-19 pandemic, (5) ownership of school equipment, (6) the ability to pay for school, (7) participation in formal and non-formal skills training and (8) knowledge in agriculture and non-agriculture.

2.  Health

The health dimension consists of six indicators, including (1) the number of people with mild illness in the farmer's household, (2) the number of seriously ill patients in the farmer's household, (3) the number of household members who lost their jobs due to illness, (4) the number of family members who did not follow social health insurance, (5) the quality of health services that were obtained by members of the farm household and (6) the ability to pay for health care.

3.  Living conditions

The dimension of living conditions has nine indicators, including (1) the availability of water sources for drinking, bathing, washing and cooking; (2) water treatment before consumption; (3) access to obtain food stuffs; (4) pet cage conditions; (5) toilet conditions; (6) household waste processing; (7) residential road conditions; (8) source of lighting and (9) cooking fuel used by farm households.

The Multidimensional Poverty Index (MPI) is measured by using weight of dimensions and measurement indicators. The weight of each dimension is weighed the same, namely 1/3. The weight of each indicator from each dimension is also weighed the same. The education dimension consists of 8 indicators rated at 1/24. The health dimension consists of 6 indicators rated at 1/18. The dimension of living conditions consists of 9 indicators rated at 1/27. The poverty assessment of each person per each indicator has a range of 0–1. Point 1 is awarded to people who meet the poverty criteria. Then, it is calculated by the following formula:

$$C_i = W_1 I_1 + W_2 I_2 + \ldots + W_d I_d \tag{1}$$

$$H = \frac{q}{n} \tag{2}$$

$$A = \frac{\sum_{i=1}^{n} ci\,(k)}{q} \tag{3}$$

$$MPI = H \times A \tag{4}$$

$Ii$ = 1 if the respondent is poor in indicator *i*, and $Ii$ = 0 if the respondent is not poor in indicator *i*. *wi* is the weight of indicator *i* with $\sum_{i=1}^{d} wi = 1$. All indicators and dimensions are summed, and the average value is sought. If the total average is greater than 1/3, the respondent was categorized as poor. H is the multidimensional headcount ratio, and A is the intensity of poverty. *q* is the number of individuals categorized as multidimensional poor, and *n* is the total population. $c(k)$ is the score of individual id, and *q* is the number of individuals who experience multidimensional poverty. It is assumed that the model will fit with the observed data.

**Table 2.** Dimension and Indicators of Multidimensional Poverty of Transmigrant and Local Household.

| Dimensions of Poverty | Indicator Modification | Indicators Cut Offs | Score |
|---|---|---|---|
| Education, (1/3) | Elementary school education (SD) of the head of the family | If the head of the family finished more than 6 years of elementary school or has dropped out of elementary school | 1/24 |
| | Junior high school education (SMP) of the head of the family | If the head of the family finished more than 3 years of junior high school (SMP) or has dropped out of junior high school | 1/24 |
| | Children's education | If any children in the family ages 7–16 years are not in school | 1/24 |
| | Children's participation in education during the COVID-19 pandemic | If, during the COVID-19 pandemic, there are children not in school | 1/24 |
| | Children's school equipment | If any children do not have school equipment | 1/24 |
| | Ability to pay school fees | If the head of the family cannot pay for the child's school fees | 1/24 |
| | Participation in training | If skills training was never attended | 1/24 |
| | Knowledge possession | If there is lack of knowledge in any field | 1/24 |
| Health (1/3) | The amount of family members who are mildly ill | If there is a family member with a mild illness | 1/18 |
| | Number of healthy family members (severe illness) | If there is someone in the family who is seriously ill | 1/18 |
| | Lost job due to illness | If a family member loses his job due to illness | 1/18 |
| | Participation in health social security/health insurance | If there are family members that do not have joint health insurance | 1/18 |
| | Health services | If family members do not have good health services | 1/18 |
| | Ability to pay for health services | If the family cannot pay for health services | 1/18 |
| Living Conditions (1/3) | Water sources | If the household has difficulty obtaining a source of drinking water | 1/27 |
| | Water condition | If the household does not cook the water before it is consumed | 1/27 |
| | Food source | If the household has difficulty obtaining and lacks food sources | 1/27 |
| | Farm conditions (cattle) | If the cattle cage is combined with the home | 1/27 |
| | Toilet condition | If the household does not own a toilet | 1/27 |
| | Sanitary conditions | If the household does not have a place to dispose of daily garbage | 1/27 |
| | Road conditions | If the road condition infrastructure has not been cast or is not asphalt | 1/27 |
| | Home lighting conditions | If the household does not have lighting at night | 1/27 |
| | Type of fuels used | If the household still uses wood fuel for cooking | 1/27 |

Logistic regression method is used with the following equation:

$$Z_P = \alpha + \beta_1 X_1 + \beta_2 X_2 + \beta_3 X_3 \ldots + \beta_n X_n + U_i \tag{5}$$

Equation (5) consists of 6 independent variables. The explanation of the above equation is shown in Table 3.

**Table 3.** Variables and expected correlation with poverty.

| Variable | Variable Description | Expected Correlation Sign |
|---|---|---|
| | Dependent variable | |
| Poverty status ($Z_p$) log odds dependent variable | Not poor = 0, poor = 1 | |
| | Independent variable | |
| Age ($X_1$) | Total age | + |
| Education ($X_2$) | No school = 0 Primary school = 1, Junior high school = 2, Senior high school = 3, College = 4 | - |
| Household size ($X_3$) | Number of people | - |
| Participation in training ($X_4$) | 0 = not attending training 1 = join training | - |
| Side business ownership ($X_5$) | 0 = no side business 1 = have a side business | - |
| Access to credit ($X_6$) | 0 = not accessible 1 = accessible | + |

Based on the formulated variables above, it is assumed that the model fits with observational data. A separate logistic regression was performed to determine the effect of the dummy variable based on the time of transmigration placement and the type of household group. The time of transmigration placement consists of initial placement, medieval placement dummy and new placement dummy. The placement dummy variable is used as a comparison variable so that there will be two dummies in the logistic analysis, namely the initial placement dummy and the medieval placement dummy. As for the household group dummy, the local household dummy is used as a comparison variable so that one dummy will appear in the analysis results, namely the transmigrant dummy.

$$Z = \alpha + \beta_1 D_1 + \beta_2 D_2 + \ldots + \beta_n D_n + Ui \qquad (6)$$

In Equation (6) there is 1 transmigrant dummy variable and 2 initial and medieval placement dummy variables. Table 4 explains the description of the dummy variable and the expected correlation sign from Equation (6).

**Table 4.** Variables and expected correlations with poverty.

| Variable | Variable Description | Expected Correlation Sign |
|---|---|---|
| | Dependent variable | |
| Poverty status (z) log odds dependent variable | Poor = 1, Others = 0 | |
| | Independent variable | |
| Dummy transmigrant ($D_1$) | Transmigrant = 1 Local = 0 | - |
| Dummy initial placement ($D_2$) | Initial placement = 1 Others placement = 0 | - |
| Dummy intermediate placement ($D_3$) | Intermediate placement = 1 Others placement = 0 | - |

## 3. Results

### 3.1. Research Location

Gorontalo and Boalemo Districts, as the location of this research, are two regional government areas in Gorontalo Province. Based on the geographical position shown in Figure 2, it is explained that Boalemo District in the north is bordered by North Gorontalo District, in the south by Tomini Bay, in the west by Pohuwato District and in the east by Gorontalo District.

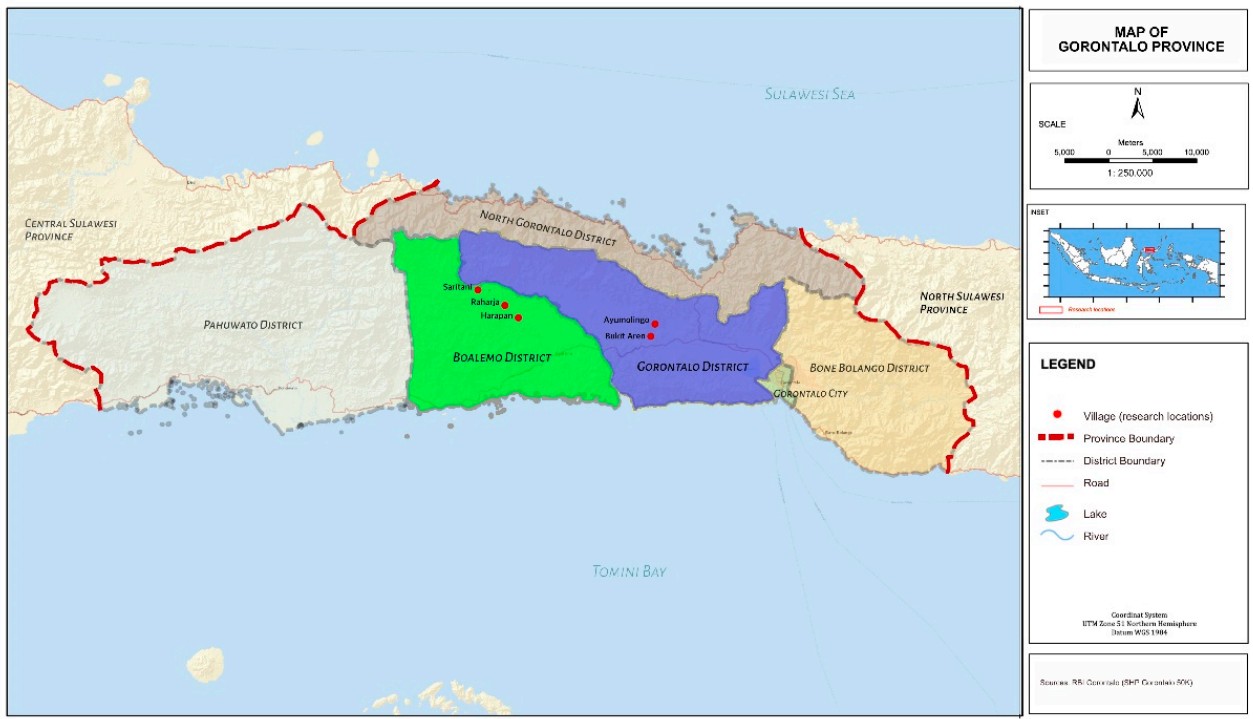

**Figure 2.** Location of five villages as research site in Boalemo and Gorontalo district map of Gorontalo province, Indonesia.

Based on the BPS data (2020), the population of Boalemo District was 145,868 people; the majority worked as laborers/employees/government officers (22,220 people) and temporary workers (20,023). Boalemo consists of Gorontalo natives and transmigrants who came from East Java and Bali through the transmigration program in 1976–1981 and settled in the villages of Raharja and Harapan. On this basis, Raharja and Harapan villages are categorized as the initial placement group, while Saritani village is categorized as the new placement group, because in 2019 only transmigrant households were placed. Agriculture and livestock are the main sources of income for transmigrant households by relying on chili, corn, paddy rice and beef cattle as commodities. The economic activities of transmigrant households are also engaged in trading as a source of side income.

Based on the results of the population census in 2020, the population of Gorontalo District is 393,107 people. The people of Gorontalo District consist of local Gorontalo natives and transmigrants from outside Gorontalo who came to Gorontalo through the transmigration program. The transmigration household group in Ayumolingo and Bukit Aren villages was categorized as the medieval placement, due to their move starting in 2016, and came from East Java, Central Java, West Java, Yogyakarta and Banten. The pattern of transmigration development in Gorontalo district is a dry land that relies on the agricultural sector as the main source of income rather than as laborers, employees or government officials (32,368 people). In the agricultural sector, corn is the main product of Gorontalo District, with a harvested area of 73,888.0 ha. Gorontalo District is also the largest vegetable crop producing area in Gorontalo Province, growing food such as chilies, cayenne peppers, tomatoes, onions and cucumbers.

### 3.2. Characteristics of Transmigrant and Local Households

Education, household size, gender and age of the head of the household are significantly associated with poverty status (Shaukat et al. 2019; Kedir and Sookram 2011). Table 5 describes the detailed conditions regarding the characteristics of age, education, household size and education level of 240 respondents.

**Table 5.** Characteristics of research respondents (N = 240) (%).

| Item | | Household (%) | |
|---|---|---|---|
| | | **Transmigrant** | **Local** |
| Age | <27 | 3 (2.50) | 25 (20.83) |
| | 27–37 | 21 (17.50) | 36 (30.00) |
| | 38–48 | 40 (33.33) | 31 (25.83) |
| | 49–59 | 41 (34.17) | 18 (15.00) |
| | 60–70 | 13 (10.83) | 9 (7.50) |
| | >70 | 2 (1.67) | 1 (0.83) |
| Gender | Male | 83 (69.17) | 72 (60.00) |
| | Female | 37 (30.83) | 48 (40.00) |
| Household Size | 1 | 4 (3.33) | 1 (0.83) |
| | 2 | 25 (20.83) | 25 (20.83) |
| | 3 | 28 (23.33) | 37 (30.83) |
| | 4 | 40 (33.33) | 43 (35.83) |
| | 5 | 14 (11.67) | 13 (10.83) |
| | 6 | 8 (6.67) | 0 (0.00) |
| | 7 | 1 (0.83) | 1 (0.83) |
| Education | No school | 2 (1.67) | 3 (2.50) |
| | Elementary School | 63 (52.50) | 107 (89.17) |
| | Junior High School | 20 (16.67) | 6 (5.00) |
| | High School | 34 (28.33) | 4 (3.33) |
| | College | 1 (0.83) | 0 (0.00) |

Source: Primary data 2021, the lowest level of education is elementary school while the highest level of education is college. Household size is the number of family members living in one house. Productive age range is 15–64 years. The data show the number of respondents in each item followed by percentage.

Age was an important component in the discussion of poverty calculations (Direja 2021). Table 5 explains that, by age, local households are more millennial farmers than transmigrant households. This is because transmigration participants are dominated by an older age. This condition needs to be reconsidered by the government to include millennial groups as transmigration participants because millennial farmers have good technical and managerial competencies (Haryanto et al. 2022).

The gender of the research respondents was dominated by men. In research (Sugiyono and Sriningsih 2018), gender has an influence on the probability of a household being poor. The agricultural work is dominated by males because farming activities require more power, particularly planting and harvesting. Not only that, the severe conditions of the transmigration area and the farming land, which are far from the residence, make men considered more likely to survive in transmigrant areas than women.

The role of the family is very important in the agricultural sector as a source of labour (Baliyan 2018) as well as a decision maker (Salahuddin et al. 2017). Table 5 explains that most transmigrant and local households consist of four members. Family size can be seen from two sides. First, the larger the size of the family, the greater the number of dependents in the family. This condition has an impact on higher household expenditures, both in the form of consumption and education costs. In research (Hanum 2018), it was explained that the number of dependents has a negative and significant influence on the income of poor families. This is in line with (Borko 2017), which states that the severity of poverty in rural areas was triggered by the family size. On the other hand, in several studies, the number of dependents in the family has a significant negative relationship with the poverty level,

because more and more household members are working, so that the source of income that comes from other family members is family savings (Shaukat et al. 2019).

Based on the condition of education level, Table 5 explains that the majority of transmigrant and local households have elementary school education. The facts in the field also showed that from the aspect of human resource capital, transmigrant households have a better capital than local households because 28.33% have a high school education (SMA), and 16.67% have junior high school (SMP). With this educational capital, it is hoped that it will reduce the level of poverty. Research (Susanto and Pangesti 2019) states that the higher the education, the lower the probability of falling into poverty. The development of human capital depends on how the government provides guidance to improve the skills of transmigrants, so they are able to be economically independent and need to be supported by a strong spirit from the transmigrant community to change their lives for the better.

### 3.3. Multidimensional Poverty of Transmigrant and Local Households

MPI was calculated by measuring the Headcount ratio (H) and the multidimensional poverty intensity (A). Based on (Haris 2016), the headcount ratio (H) describes the percentage of multidimensional poor people, so, how many people experience multidimensional poverty in an area, while the intensity of multidimensional poverty (A) describes how many dimensions are deprived, where the greater the value of A indicates the more dimensions that are deprived. This condition means that the problem of poverty in the area was getting more complex. In the analysis of multidimensional poverty calculations in this study, there are three groups based on the year of placement of the transmigration program, namely the initial (1976), medieval (2016) and new placement groups (2019).

Table 6 describes that the poor transmigrant households in new placements (0.869) are higher than transmigrant households in medieval (0.726) and old (0.478) placements. This happens because the transmigrant households have not managed the two types of farming land provided by the government (farmland 1 and 2) to the maximum, because the distance between the farmland is far from the location of residence, and some farmland 2 is not ready to be used for agricultural activities. The headcount ratio value of transmigrant households was smaller than that of local households in both the old (0.478 < 0.791) and new (0.869 < 0.956) placements. This condition exhibits that the transmigration program is able to lower poverty.

**Table 6.** Multidimensional Poverty Index (MPI) by year of placement (%).

| Transmigration Placement Location | Placement | Village | Multidimensional Headcount Ratio (H) | | Intensity of Poverty (A) | | Multidimensional Poverty Index (MPI) | |
|---|---|---|---|---|---|---|---|---|
| | | | Trans | Local | Trans | Local | Trans | Local |
| Bongo I dan II | Initial | Raharja dan Harapan | 0.478 | 0.791 | 0.460 | 0.486 | 0.220 | 0.384 |
| Ayumolingo and Bukit Aren | Medieval | Ayumolingo dan Bukit Aren | 0.726 | 0.712 | 0.435 | 0.419 | 0.316 | 0.299 |
| Pangea SP 3 | New | Saritani | 0.869 | 0.956 | 0.471 | 0.519 | 0.409 | 0.497 |

The magnitude of the poverty intensity is described in Table 6. Transmigrant households with new placements (0.471) contribute a higher intensity of poverty value than the initial placements (0.460). The results of this analysis indicate the condition that the poverty dimension among transmigrant households with new placements is more deprived than households with old placements. The same condition is also shown in the value of the intensity of poverty of transmigrant households, which is smaller than that of local households in both the initial (0.460 < 0.486) and new (0.471 < 0.519) placements. These results explain that the transmigration program is able to reduce the risk of being deprived in every dimension of poverty, which includes the dimensions of education, health and livelihoods.

The calculation of the Multidimensional Poverty Index (MPI) based on Table 6 explains that the initial placement (0.220) transmigrant households have a lower MPI value than the medieval (0.316) and new (0.409) placements. Higher poverty numbers in the transmigration area for new placements is triggered by infrastructure facilities, such as road access at several transmigrant locations, which are still very poor, one of which is in Saritani village, because this village is a transmigrant area that was recently opened by the government. The distance between villages and economic centres such as markets is also very far, so transportation and transportation costs are higher. Another reason is that transmigrants' and local farmers' income sources mostly rely on corn farming and do not have other sources of income. Likewise, some of the farmland 2 is not ready to be cultivated as a means of farming. Table 6 also explains that the MPI value of initial (0.220 < 0.384) and new (0.409 < 0.497) transmigrant houses is lower than that of local households. The MPI value of transmigrant households, which is smaller than local, also indicates that the transmigration program is able to reduce the chances of farmer households remaining poor or falling into poverty in the future.

Based on the results analysis in Table 6, there are new findings that explain the longer the transmigration program lasts, the more opportunities there are for transmigration households to get out from poverty. Transmigration placements that have been running for 35 years show a much lower MPI value when compared with new placements. This condition is supported by better road infrastructure, the availability of economic facilities such as banks and markets, educational facilities that are easily accessible, and the social life of the community between transmigrant and local households running in a sustainable manner. Based on the results of the analysis, it can be used as a recommendation for the government that the transmigration program is able to have an impact on reducing poverty in the transmigration area of Gorontalo Province.

*3.4. Poverty Conditions from the Dimensions of Education, Health and Livelihood*

Table 7 describes each indicator on each deprived dimension. In the education dimension, indicators of knowledge and skill level in the agricultural sector and other sectors are the most deprived poverty indicators for transmigrant and local households, whereas having skills and knowledge is important to help farmer households get out of poverty (Lahat 2017; Ilemona et al. 2013). In addition, the limited ability to pay school fees is also an indicator of quite high deprivation of poverty in transmigrant and local households, with a range of 8.06–17.12%. Even though they are in poor condition, from the completeness aspect of school facilities such as stationery, they are in a good condition.

Based on the health dimensions in Table 7, health service indicators are the most deprived indicators for transmigrant and local households, because poverty causes the inability to pay for health facilities. This shows that poverty results in powerlessness in obtaining better health services; this result is in line with research (Peters et al. 2008; Butler 2006; Lazar and Davenport 2018). The government has made efforts through the provision of national health insurance, but community participation in this program is still low and even fewer rural residents access this facility (Nugraheni and Hartono 2017), whereas participation in health insurance provides a greater opportunity to access clinics and other health facilities (Allison et al. 2016).

The dimension of living conditions shows that the condition of infrastructure (35.90%) is the biggest cause of poverty in transmigration areas, so that transportation expenditures burden transmigrant and local households who are categorized as poor households, whereas transportation is needed to access economic activities such as markets. Inadequate infrastructure reduces access to opportunities that can lift them out of poverty (Blumenberg and Agrawal 2014). Another indicator that is still a limitation of transmigrant and local households is in terms of processing household waste. The composition of transmigrant household waste is dominated by yard waste, food waste and plastic, which are simply thrown away without any use, for example as fertilizer. There is also no place for household waste to be disposed of, even just dumped carelessly close to the house. Limited land

for landfills and the high cost of collecting and transporting waste is a challenge in waste management itself (Maryanti 2017). Indicators of the condition of livestock pens are also a problem for transmigrant and local households. The cattle barn is still one with the residence. Several studies have stated that ownership of livestock as food reserves reduces poverty and is a capital asset for the rural poor, which is usually inherited from the family (Flores et al. 2020; Agus and Widi 2018). Indicators that are considered good are the lighting conditions that are already owned by transmigrant and local households.

**Table 7.** Dimensions and indicators of multidimensional poverty measurement (%).

| Dimension | Indicator | Raharja and Harapan | | Ayumolingo and Bukit Aren | | Saritani | |
|---|---|---|---|---|---|---|---|
| | | Trans | Local | Trans | Local | Trans | Local |
| Education | Elementary school | 9.71 | 5.43 | 5.45 | 6.76 | 8.06 | 10.86 |
| | Junior high school | 14.56 | 12.40 | 11.26 | 14.42 | 16.94 | 16.00 |
| | Number of school children | 11.65 | 6.98 | 8.26 | 9.91 | 8.87 | 10.86 |
| | School participation during the pandemic | 4.85 | 13.95 | 6.21 | 5.86 | 4.84 | 10.29 |
| | Ownership of school supplies | 4.85 | 8.53 | 4.41 | 5.41 | 4.84 | 10.29 |
| | Ability to pay school fees | 10.68 | 15.50 | 17.03 | 13.97 | 8.06 | 10.29 |
| | Participation in skills training | 21.36 | 17.83 | 17.59 | 17.12 | 24.19 | 16.57 |
| | Possession of knowledge and skills | 22.33 | 19.38 | 29.81 | 26.58 | 24.19 | 14.86 |
| Health | Patients with mild pain | 27.91 | 18.35 | 29.55 | 28.63 | 24.27 | 28.89 |
| | Seriously ill patient | 2.33 | 2.75 | 3.72 | 0.64 | 1.94 | 0.00 |
| | Lost job due to illness | 5.81 | 5.50 | 2.15 | 4.32 | 0.00 | 1.11 |
| | Health insurance participation | 22.09 | 23.85 | 10.25 | 16.86 | 20.39 | 20.00 |
| | Quality of health services | 12.79 | 22.94 | 22.05 | 21.01 | 25.24 | 23.33 |
| | Ability to pay for health facilities | 29.07 | 26.61 | 32.29 | 28.56 | 28.16 | 26.67 |
| Living conditions | Water access | 4.76 | 0.00 | 1.75 | 1.30 | 0.00 | 4.44 |
| | Drinking water safety | 40.48 | 2.50 | 5.82 | 4.08 | 3.85 | 8.89 |
| | Food source | 4.76 | 17.50 | 9.30 | 33.32 | 3.85 | 2.22 |
| | Condition of the cattle barn | 23.81 | 20.00 | 12.21 | 4.55 | 6.41 | 15.56 |
| | Toilet sanitation | 2.38 | 2.50 | 9.30 | 8.34 | 11.54 | 10.00 |
| | Household waste management | 4.76 | 5.00 | 13.37 | 18.08 | 29.49 | 25.56 |
| | Road conditions | 19.05 | 35.00 | 33.14 | 24.10 | 35.90 | 32.22 |
| | Source of light | 0.00 | 12.50 | 1.16 | 0.65 | 0.00 | 0.00 |
| | Types of household fuel | 0.00 | 5.00 | 13.96 | 5.61 | 8.97 | 1.11 |

A summary of the results of analysis related to poverty in each dimension is shown in Table 8. The findings explain that the level of deprivation of transmigrant households in the initial placement is smaller than the new placement in the dimensions of education, health and livelihoods. This condition indicates that transmigrant households in the initial placement have obtained access to education, health and a better life than transmigrant households with new placements, including road access, economic facilities such as markets and school education facilities. Table 8 also explains that the health dimension is the most deprived component of the poverty dimension in transmigrant households. Problems related to health include the number of family members who suffer from illness is still very high and the inability to pay for health facilities.

Different conditions occur in local households, namely the education dimension as the dimension with the highest level of deprivation compared to the health and livelihood dimensions. Prominent poverty problems in local households include the lack of heads and household members in accessing formal and non-formal education. Most of the respondents' education only finished at elementary school. In non-formal education, it is still rare for family members to participate in skills training to increase knowledge as a capital to be economically independent. Based on Table 8, it is also known that the level of deprivation of transmigrant households in the initial placement is smaller than the new placement in the dimensions of education, health and livelihood. This fact can be the

basis for the formulation of government policies that health and education issues are still becoming a big agenda for the Gorontalo Provincial government.

**Table 8.** Poverty rates for each MPI dimension.

| Transmigration Placement Location | Placement | Village | Education | | Health | | Living Condition | |
|---|---|---|---|---|---|---|---|---|
| | | | Trans | Local | Trans | Local | Trans | Local |
| Bongo I and II | Initial | Raharja and Harapan | 4.29 | 5.38 | 4.78 | 6.06 | 1.56 | 1.48 |
| Ayumolingo and Bukit Aren | Medieval | Ayumolingo and Bukit Aren | 3.63 | 4.63 | 5.17 | 4.48 | 3.19 | 2.30 |
| Pangea SP 3 | New | Saritani | 5.17 | 7.29 | 5.72 | 5.00 | 2.89 | 3.33 |

*3.5. The Influence of Other Factors toward Poverty of Transmigrant and Local Households*

When measuring MPI (Multidimensional Poverty Index) apart from the dimensions of education, health and living conditions, it is suspected that there are other factors that affect the poverty level. In the study (Shaukat et al. 2019; Meyer and Niyimbanira 2016), it was explained that the factors of household size, gender and age of the head of the household were significantly related to poverty status. Likewise, skills have a high correlation with job opportunities and income (Foundation 2016). Possession of skills will provide wider job opportunities and will ultimately support the family by increasing household income and economic growth. Improved economic conditions will tend to reduce poverty. Research (Naminse et al. 2019) found that there is a significant and positive relationship between entrepreneurship of poor households in poverty alleviation. Household members should combine the efforts of multiple livelihood strategies, such as food and cash crop production, wage labour in agriculture or enterprises and entrepreneurial activities of micro and small enterprises (Vandenberg 2006). From the aspect of financial credit, it also shows that it is effective in helping to reduce multidimensional poverty (Khaki and Sangmi 2017), although research (Ampah et al. 2017) shows that access to credit and financial services has a weak positive effect on income growth, increased consumption expenditure and acquisition of business assets, it has a significant effect on education of children in poor households, and education is one indicator of poverty measurement.

Based on the results of the analysis in Table 9, the chi square value in the Hosmer and Lemeshow test is 12,580 with a significance of 0.127 > 0.05, so Ho is accepted and Hi is rejected. This means that the model formed fits with observational data. Other results show that the overall percentage correct value of 79.6 means that the binary logistic model produces 79.6% accuracy of poverty prediction. The value of the omnibus tests of model coefficients is equivalent to the F-test in multiple linear regression (Latan 2014). Based on the results of the analysis above, it rejects the null hypothesis because the significance of the model is 0.001 < 0.05, so it can be concluded that the independent variables consisting of age, education, household size, skills training participation, side business ownership and access to credit together have a significant effect on predicting poverty rates in Gorontalo Province.

The magnitude of the effect is indicated by the value of the odds ratio Exp (B), a value of B above 1.0 indicates a positive effect, and a value below 1.0 indicates a negative effect (Latan 2014). The age variable with an odds ratio value of B 1.014 > 1.0 means that farmer households with an older household head are at greater risk of having a poor life than younger household heads. The results of this study are in line with research (UNDESA 2015), which states that getting older has an additional risk of remaining poor or becoming poor. The tendency of respondents who are older will reduce working hours or stop working because of health problems or deliberately stop working as farmers. If the old head of the household decides to continue working, it is usually as a farm labourer or factory worker with a small wage. The average farmer household does not have savings for old age. Some of them sell their land in their old age to meet their daily needs and to meet

medical expenses. Based on this fact, the transmigration program needs attention about age of transmigrant participants as an effort to regenerate young farmers in Indonesia.

**Table 9.** Logistic regression of the effect of other selected factors on poverty.

| Independent Variables | B | Odds Ratio Exp (B) | Sig |
|---|---|---|---|
| Constant | 3.197 | 24.463 | 0.003 |
| Age | 0.013 | 1.014 | 0.386 |
| Education | −0.852 | 0.426 | 0.001 |
| Household size | 0.065 | 1.067 | 0.672 |
| Skills training participation | −1.007 | 0.365 | 0.004 |
| Side business ownership | −0.922 | 0.398 | 0.031 |
| Access to credit | −0.300 | 0.741 | 0.436 |
| Hosmer and Lemeshow test/Chi square | | 12.580 | 0.127 |
| Overall percentage correct | | 79.6 | |
| Omnibus tests of model coefficients | | 40.586 | 0.001 |

Source. Primary data is processed.

Different conditions are indicated by the education variable with an odds ratio Exp (B) of 0.426 < 1.0; this value indicates that education has a negative and significant effect of 0.001 < 0.05 on the poverty level. Farm households with low education are more likely to remain poor or become poor than farm households with higher education. If the respondent does not go to school or only has elementary school education, then the household has a 42.6% chance to remain poor and even become poor in the future. This result is reinforced by research (Shaukat et al. 2019), which states that higher education has the opportunity to reduce poverty levels, such as poor farmer households with low education if they have a side job, they work in other informal sectors and are only capable of being farm laborers or small industrial workers. Therefore, efforts are needed not only to provide job skills but also to establish a minimum wage for the informal sector in rural areas (Kathuria and Raj S. N. 2016).

Table 9 explains that household size has a positive effect on farmer household poverty. The household size variable with a B value of 1.067 interprets that households with bigger household sizes are 1067 times more likely to experience poverty than households with a few family members. In the agricultural sector, household members play a role not only as workers; even farming decisions sometimes depend on decisions by their family. In the study (Fusco and Islam 2017; Meyer and Niyimbanira 2016), it was also stated that the number of children from different age groups significantly affects poverty.

The variables of participation in skills training referred to in this study are farming training and various non-agricultural skills training which are expected to be able to provide economic independence for transmigrant and local farmers' households. The value of the odds ratio Exp (B) of the skills training participation variable is 0.365 and significant at 0.004 < 0.05; this value suggests that if the farmer's household has never participated in skills training, 39.8% is at risk of remaining poor or becoming poor. The results of this study are in line with (King and Palmer 2006; Foundation 2016), which states that there is a significant correlation between skills and employment opportunities with poverty reduction, and that knowledge and skills can be obtained through formal and non-formal learning activities. It should be noted that the skills training program must be right on target; in this case, the skills are adjusted to the potential of the transmigration destination area. Not only that, it is also necessary to pay attention to the interests and talents of transmigrant participants, so that by participating in skills training they will have the knowledge to find and maintain jobs and will have an impact on increasing farmers' household income. The skills needed are also useful for dealing with and adapting to the uncertainties of change in the labour market (Foundation 2016).

In several studies on entrepreneurship of poor households, it was found that entrepreneurship is considered capable of being a solution to poverty reduction (Shepherd

et al. 2020; Naminse et al. 2019). In Table 9, the value of the odds ratio Exp (B) of the side business ownership variable is 0.398, which means it is significant at the value of 0.031 < 0.05; this value means that the farmer household who has a side business has a 39.8% opportunity to reduce poverty and even get out of poverty. Farmers' household side income can be as farm laborers, factory workers, livestock businesses and aquaculture (Adriansyah et al. 2020). Although research (Setyawati 2019) shows that there are various obstacles to entrepreneurship development in the transmigration area, including, among others, the business being run is not focused, knowledge diffusion, weak skills, has not been connected to entrepreneurship development programs in other sectors or technical units and there has been no change in the way of thinking in formulating business development needs. It is necessary to believe that poor household communities also have an entrepreneurial orientation, only that the level of orientation differs based on experience, education and income (Kumar et al. 2018). Various programs were launched by the government to grow micro, small and medium enterprises businesses, including the Joint Business Group (KUBE) and Productive Economic Enterprises (UEP). The aim of various business programs for poor communities is to improve the livelihoods of vulnerable business communities through providing relevant skills and helping access the financing needed to increase income opportunities for poor farmer households.

The variable access to credit has an odds ratio Exp (B) of 0.741; this result shows that if farmer households can access credit, then 74.1% have the opportunity to reduce the risk of farmer households remaining poor or falling into poverty. The results of this study are in line with (Khaki and Sangmi 2017; Rozanti et al. 2021). Credit access can usually be obtained from formal institutions such as banks or cooperatives, but the farming household community obtains access to credit from non-formal institutions. Most of them get credit from their close friends and family, even from rice mill owners who give loans on condition that after harvesting, the products are sold to rice mill owners. This variable is interesting because credit is used for productive activities or to meet cost for daily life. There is a need for further studies regarding this variable.

According to the information from the interview, it was stated that the credit was used more for education expenses. So it could be that the credit obtained is only able to solve one of the many problems of poverty such as education. Likewise, research (Ampah et al. 2017) states that access to credit and financial services has a low effect on income growth, increased consumption expenditure and asset acquisition, but has a significant effect on the ability to educate children. In another study, access to credit was termed as financial inclusion. The level of poverty and income inequality in developing countries is significantly affected by financial inclusion (Omar and Inaba 2020). Based on the results of the analysis of this study and supported by various references, the use of formal financial services for groups of poor farmer households needs to be increased in order to maximize the welfare of the community as a whole.

The chi-squared value in the Hosmer and Lemeshow test was 3.002, with a significance of 0.809 > 0.05, so Ho was accepted and Hi was rejected. This means that the model formed fits the observational data. The overall percentage correct value of 75.8 means that the overall prediction accuracy of the model is 75.8%. The omnibus test value is the capability of the predictor in the model to predict the response of variable Y. The omnibus test value sig 0.001 < 0.05 means that the transmigration and placement dummy together have a significant effect on predicting the poverty level in Gorontalo Province.

Table 10 explains that the transmigrant dummy variable has an odds ratio Exp (B) value of 0.551; this result indicates that if a farmer household does not participate in the transmigration program, 55.1% is at risk of remaining poor or becoming poor in the future, or transmigrant households have a lower poverty value by 0.596 times compared to local households, although it must be acknowledged that there are still several factors that must be addressed in the implementation of the transmigration program, including land ownership status forming legal ownership of yard land, farmland 1 and farmland 2, road infrastructure condition and communication network. The result of analysis indicates

that the transmigration program has the opportunity to reduce poverty. This is the basis for policy formulation that the transmigration program can be continued with some improvements for the successful implementation of the program. Improvements that can be made include the legality of ownership of yards and farmland, road infrastructure and economic activity facilities such as markets, training on business skills development and entrepreneurship to train economic independence.

**Table 10.** Logistic regression of the effect of transmigration and its time of placement on poverty.

| Independent Variables | B | Odds Ratio Exp (B) | Sig |
|---|---|---|---|
| Dummy transmigrants | −0.596 | 0.551 | 0.062 |
| Dummy of early/initial placement | −2.352 | 0.095 | 0.001 |
| Dummy of medieval placement | −2.199 | 0.111 | 0.001 |
| Constant | 3.282 | 26.631 | 0.001 |
| Hosmer and Lemeshow test/Chi square | | 3.002 | 0.809 |
| Overall percentage correct | | 75.8 | |
| Omnibus tests of model coefficients | | 25.077 | 0.001 |

Source. primary data processed.

In the next analysis, groupings are made based on the time of transmigration placement, consisting of initial, medieval and new placements. The analysis was conducted by looking for the effect of transmigration placement on poverty status. This study obtained a new finding that the transmigration program has a significant effect on the poverty status of farmer households. Dummy initial placement of the transmigration program has a significant effect on poverty status $\alpha$ = 0.05. The results of the odds ratio value of 0.095 show a negative effect or with a value of B = −2.352, which means that households who participate in the initial placement transmigrant program and have been running for 32 years have a 2.352 times chance for not being poor now and in the future.

The results of the analysis also explain that the medieval placement dummy has a significant effect on the poverty status of transmigrant households with a level $\alpha$ = 0.05 and the odds ratio value of Exp 0.111, which means that the medieval placement dummy has a negative effect on the poverty status of transmigrant households. The B value of −2199 explained that households participating in the transmigration program for medieval placement have a 2199 times chance of reducing poverty now and in the future. This indicates that a longer implementation of the transmigration program provides the opportunity to avoid a high level of poverty in farmer households. This result is one proof that the implementation of the transmigration program is to be able to reduce the level of poverty in the transmigration area of Gorontalo Province.

## 4. Conclusions

a. The multidimensional poverty rate of transmigrant households is lower than that of local households. The longer the placement of transmigration, the more likely it is to reduce the current and future poverty level of transmigrant households.

b. The health dimension is the most deprived aspect compared to the education dimension and living conditions in transmigrant households. The limited ability to pay for health facilities, the high number of people with mild illnesses and limited access to quality health services are three indicators of multidimensional poverty in transmigrant households. However, in local households, the education dimension has the highest level of deprivation. The limitation of formal and non-formal education is a major poverty problem of local households in Gorontalo Province.

c. Logistic regression analysis proves that education, skills training participation and side business ownership have a negative and significant effect on multidimensional poverty of transmigrant and local households; however, age and household size have a non-significant (+) effect on poverty, and access to credit has a non-significant effect (−) on the poverty of transmigrant and local households in Gorontalo Province.

d. The results of the analysis show that the transmigration program provides a significant opportunity in reducing poverty; this is evidenced by the results of the analysis, which show that initial placement transmigrant households are 2352 times more likely to be non-poor now and in the future compared to medieval and new placement transmigration.

**Author Contributions:** Conceptualization, I.I. and A.M.; methodology, I.I. and A.M.; validation, J.H.M.; formal analysis, I.I., A.M., and L.R.W.; investigation, J.H.M. and L.R.W.; data curation, J.H.M. and L.R.W.; writing original draft preparation, A.M. and I.I.; writing review and editing, I.I and A.M.; supervision, I.I., A.M., J.H.M and L.R.W. All authors have read and agreed to the published version of the manuscript.

**Funding:** This research and APC was funded by Indonesia Endowment Fund for Education (LPDP) Ministry of Finance Republik Indonesia. (Lembaga Pengelola Dana Pendidikan Republik Indonesia). Grand number is KET-140/LPDP.4/2020 and KET-2131/LPDP.4/2022.

**Institutional Review Board Statement:** Not applicable.

**Informed Consent Statement:** Not applicable.

**Data Availability Statement:** Data is available upon request.

**Acknowledgments:** The authors convey a sincere thanks and appreciation to Dinas Tenaga Kerja and Transmigrasi of Gorontalo district and Dinas Tenaga Kerja dan Transmigrasi of Gorontalo district for assisting field work.

**Conflicts of Interest:** The authors declare no conflict of interest.

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
