# Peer review of "The Effects of the Transmigration Programme on Poverty Reduction in Indonesia’s Gorontalo Province: A Multidimensional Approach"

_economies, doi:10.3390/economies10110267_

Round 1

Reviewer 1 Report

In my opinion, the article entitled “The Effects of the Transmigration Program on Poverty Reduction in Indonesia’s Gorontalo Province: A Multidimensional Approach” is very interesting. It covers a very crucial area of scientific research. However, some aspects of this article must be rethought:

1)     The authors have written (lines 58-59) that “the new thing that is shown in the research is the inclusion of new indicators on the dimensions of education, health and living conditions”. But a few lines earlier it is written that “…with the Alkire Foster method approach, there are at least three measurement dimensions used, namely education, health and living standards”.  So the authors should clearly explain what exactly new things they bring to science.  In the presented categories they seem exactly the same as other authors.      

2)     I propose that in table 5, where the characteristic of research respondents is presented, the number of people in the households should be given and the household size could be put in parentheses.

3)      I have some objections as to the methodology of the research. The Authors analyse and compare some indicators for different groups of people (initial, medieval and new placement groups) in different periods of time. The time differs: for initial and new placement groups is one year  (1987, 2019) but for medieval is a period: 2013-2017. So actually there is not a proper point of reference in the research which is normally done in indicators comparison.

I see only one base to compare: the location and the programme of poverty reduction in those districts.

4)     The research hypothesis should be created before the actual research starts, after a thorough literature revision. The authors formulated their hypothesis based on the results of the analysis presented in some tables, which is incorrect.

5)     It is advisable to point to the limits of the research.

6)     Between the lines 34-35 I propose to write: “with a gap of more than 19% between them”.

7)     In British English the word program in the article’s title should be spelt as “Programme”.

Reviewer 2 Report

The paper is relevant, and it is well developed. At the technical level, data analysis and results are sound. The conclusions are somewhat circular or expectable (I wonder if they make a substantive, factual contribution to the knowledge at stake) because they probably depart from a defined, expectable, theoretical baseline (the one I recall below). Nevertheless, I guess the paper passes the standards of a serious journal from a formal point of view – because, again, the quality, analysis, literature, and technical approaches are all remarkable.

Nevertheless, I find two main obstacles: first, a minor one, the notion/idea/concept of the "transmigration program" should be clearly explained from the outset. I suppose it is a program with its own rules and standards, but it is left unexplained.

The second obstacle, as an anthropologist, is problematic. The paper aims for a "multidimensional approach" to poverty: but it displays just one view, a very narrow economic view of poverty - by the same token, are they assuming that most peasant societies (roughly 70% of the world population) are poor? In other words, data is basically about services and material goods, but what is the role of well-being and social and emotional support, community mindset, cooperation degree, etc.? This definition of poverty assumes a globally shared standard of “middle class” in terms of material "needs," “scarcity,” or “growth" (concepts that are left unchallenged), and this is a vast and ancient topic of discussion. Formalism versus sustantivism, in Anthropology at least, displays this conflict: the main question is, can we apply the same model of Economics no matter how diverse is the sociocultural setting? In this case, the challenge is even more significant because they are comparing transmigrants and locals under the same globally accepted parameter of poverty. Let’s assume that the authors take such standards and explicitly argue/prove that the same standards can be applied anywhere in the world. Still, sociocultural differences do not count on that equation at all. This is the kind of thinking that leads to denouncing child exploitation (work done by children) in rural and traditional contexts where children are integrated into their society by helping the community and also acquiring essential knowledge and skills necessary for their living and the reproduction of their culture (TEK, transfer of knowledge from their parents or grandparents, rules of coexistence, technical skills, etc.).

My main concern was the notion of a “multidimensional approach to poverty.” This is not a multidimensional approach; it is (indeed) a very unidimensional approach to poverty, namely, an economist's perspective of poverty. There are two options here: the faster and easier to reconsider erasing all the references to such multidimensionality; or, for me, it would be more challenging and richer to struggle with a fundamental multidimensional approach where “other” categories or variables of analysis are included. In such a case, I believe authors need to expand their theoretical perspective, expose themselves to other academic challenges, and pose what is implied in a multidimensional view of poverty by reconsidering, first, if those goods and services they consider “basic needs” are, in fact, vital needs in the analyzed context (obviously, life expectancy or nutritional information are quite objective data; but what about schooling, degree of technical and professional education?). On the other hand, if aiming for a multidimensional perspective of poverty, please break disciplinary boundaries and look at different (not always radically opposed) theoretical options. I am not claiming that authors need to turn toward postmodern visions of poverty, not at all. Still, I imply that their “multidimensional perspective of poverty” should be enriched through a deeper theoretical insight.  I will drop some contemporary literature on this vast topic, both in favor and against the economist vision of poverty:

John Brohman (1995) Economism and critical silences in development studies: A theoretical critique of neoliberalism, Third World Quarterly, 16:2, 297-318, DOI: 10.1080/01436599550036149

Chang, Ha-Joon, ed. Rethinking development economics. Anthem Press, 2003.

BROAD, Robin. Research, knowledge, and the art of ‘paradigm maintenance’: the World Bank's Development Economics Vice-Presidency (DEC). Review of International Political Economy, 2006, vol. 13, no 3, p. 387-419.

Hulme, D., & Toye, J. (2006). The case for cross-disciplinary social science research on poverty, inequality, and well-being. The Journal of Development Studies42(7), 1085-1107.

Labonté, Ronald, and David Stuckler. "The rise of neoliberalism: how bad economics imperils the health and what to do about it." J Epidemiol Community Health 70.3 (2016): 312-318.

Sheppard, Eric, and Helga Leitner. "Quo vadis neoliberalism? The remaking of global capitalist governance after the Washington Consensus." Geoforum 41.2 (2010): 185-194.

Lybbert, Travis J., and Bruce Wydick. "Poverty, aspirations, and the economics of hope." Economic development and cultural change 66.4 (2018): 709-753.

Baulch, B. (Ed.). (2011). Why poverty persists: Poverty dynamics in Asia and Africa. Edward Elgar Publishing.

Mead, L. M. (2012). The poverty of poverty research. Academic Questions25(4), 539.

Bush, Ray. Poverty and neoliberalism: Persistence and reproduction in the global south. Vol. 2. London: Pluto, 2007.

Bruce, Steve. "The poverty of economism or the social limits on maximizing." Sacred markets, sacred canopies: Essays on religious markets and religious pluralism (2002): 167-185.

Mitchell, Timothy. "The work of economics: how a discipline makes its world." European Journal of Sociology/Archives Européennes de Sociologie 46.2 (2005): 297-320.

Poverty entails relationships between individuals, and individuals cannot be taken as purely economic agents because, besides sanitation, education, households, water availability, having access to meat twice a week, and salaries, they also live and relate…, and context matters.

Again, congratulations on a perfect paper that could be great if grasping/reflecting on the notion of "multidimensional poverty." 

Round 2

Reviewer 1 Report

The Authors have improved their article, according to the suggestions.

Reviewer 2 Report

Most of the considerations have been taken into account